# Identification, Fine Mapping and Application of Quantitative Trait Loci for Grain Shape Using Single-Segment Substitution Lines in Rice (*Oryza sativa* L.)

**DOI:** 10.3390/plants12040892

**Published:** 2023-02-16

**Authors:** Xiaoling Wang, Xia Li, Xin Luo, Shusheng Tang, Ting Wu, Zhiquan Wang, Zhiqin Peng, Qiyu Xia, Chuanyuan Yu, Yulong Xiao

**Affiliations:** 1National Engineering Research Center of Rice (Nanchang), Rice Research Institute, Jiangxi Academy of Agricultural Sciences, Nanchang 330200, China; 2Guangdong Provincial Key Laboratory of Plant Molecular Breeding, College of Agriculture, South China Agricultural University, Guangzhou 510642, China; 3Hainan Key Laboratory for Biosafety Monitoring and Molecular Breeding in Off-Season Reproduction Regions, Institute of Tropical Bioscience and Biotechnology & San Ya Research Institute, Chinese Academy of Tropical Agricultural Sciences, Haikou 571101, China

**Keywords:** rice, grain shape, quantitative trait locus (QTL), single-segment substitution line (SSSL)

## Abstract

Quantitative trait loci (QTLs) and HQTL (heterosis QTLs) for grain shape are two major genetic factors of grain yield and quality in rice (*Oryza sativa* L.). Although many QTLs for grain shape have been reported, only a few are applied in production. In this study, 54 QTLs for grain shape were detected on 10 chromosomes using 33 SSSLs (single-segment substitution lines) and methods of statistical genetics. Among these, 23 exhibited significant positive additive genetic effects, including some novel QTLs, among which *qTGW4-*(*1*,2), *qTGW10-*2, and *qTGW10-3* were three QTLs newly found in this study and should be paid more attention. Moreover, 26 HQTLs for grain shape were probed. Eighteen of these exhibited significant positive dominant genetic effects. Thirty-three QTLs for grain shape were further mapped using linkage analysis. Most of the QTLs for grain shape produced pleiotropic effects, which simultaneously controlled multiple appearance traits of grain shape. Linkage mapping of the F_2_ population derived from sub-single-segment substitution lines further narrowed the interval harbouring *qTGW10-3* to 75.124 kb between PSM169 and RM25753. The candidate gene was identified and could be applied to breeding applications by molecular marker-assisted selection. These identified QTLs for grain shape will offer additional insights for improving grain yield and quality in rice breeding.

## 1. Introduction

Rice (*Oryza sativa* L.), one of the world’s most important staple food crops, feeds nearly half of the global population [1]. Grain weight is a vital component of grain yield, typically evaluated based on grain shape [2,3,4,5]. With the improvement in living standards, the demand for high-quality rice has grown [6,7]. Rice grain shape, one of the most important agronomic traits, has garnered increasing attention from breeders and geneticists because it directly determines the physical appearance of grain and affects the cooking quality of rice [8,9,10,11,12,13]. The traits of grain shape are also highly heritable, rendering them valuable for genetic analyses. Therefore, yield determined by grain weight is critical to farmers because it is among the most stable components of yield to increase income [14,15]. In addition, applying heterosis loci for grain shape can also simultaneously increase yield and quality [16,17].

Grain shape development in rice has been extensively studied [18,19,20,21,22,23,24,25]. Most previous studies focused on the characterisation of mutants and expression of key genes associated with grain size, such as *Lk-f*, which confers long or short kernel size [13]. The utilisation of molecular markers has facilitated the investigation of the genetic bases of complex quantitative traits, such as grain shape. In the last decade, many independent studies on rice grain size and grain shape have been conducted in populations derived from crosses between divergent cultivars or accessions of rice [26,27,28,29]. Currently, at least 203 quantitative trait loci (QTLs) for grain shape have been detected in rice (Integrated Rice Science Database), and some QTLs or genes detected previously have been mapped and cloned in different populations with diverse backgrounds [26,30]. However, dissecting the interactions amongst QTLs is difficult with several target QTLs [1].

Furthermore, many other QTLs related to grain shape have been reported at similar locations, which typically produce minor effects on phenotypic variation [31]. Therefore, some QTLs must be validated in the population to make sure without the background effect. Single-segment substitution lines (SSSL) harbouring only a single chromosome segment are an effective material for eliminating the background effect. Thus far, nearly 10 genes have been cloned successfully, and hundreds of important QTLs have been mapped using SSSLs in rice [2,30,32,33,34,35,36,37,38]. Because of the advantages of SSSLs, more and more research has begun to apply them [39]. In this study, QTLs for grain shape were identified using 33 SSSLs. The novel grain shape loci which carry QTLs for grain yield or the heterosis loci identified in the present study will offer additional insights for studies on the fine mapping, cloning, and application of potential QTLs on rice breeding.

## 2. Results

### 2.1. Verification and Distribution of SSSLs

In 2016, 33 SSSLs harbouring 225 SSR (simple sequence repeat) and InDel (insertion/deletion) markers were identified and verified based on molecular markers (Appendix A). The substituted chromosome segments of SSSLs were distributed on 10 chromosome linkage maps, except for chromosomes 9 and 11, and the length of the substituted segments varied from 1.6–87.5 cM. Six concentrated overlapping groups, with at least three overlapping SSSLs, were present on chromosomes 1, 3, 4, 6, 7, and 8. The substitution fragments cover 36% (561/1524.9) of the whole genome (Figure 1).

### 2.2. Frequency Distribution of Phenotype in SSSLs

The distributions of the six grain shape traits of the 33 SSSLs are presented in Figure 2 (data were averaged from values collected during the early and late seasons of 2016). Values of the six grain shape traits for the receptor (HJX74) were as follows: grain length (GL) 8.12 mm, grain width (GW) 2.52 mm, length-to-width ratio (LWR) 3.24, grain size (GS) 16.91 mm^2^, grain circumference (GC) 19.40 mm, and grain roundness (GR) 0.33. Values for most SSSLs were close to those of the contrast (HJX74). In particular, the frequency distributions of grain length and grain circumference were close to normal distributions. Overall, the values of grain shape traits indicated sufficient genetic diversity (Figure 2).

### 2.3. Identification of QTLs for Grain Shape

The thousand-grain weight (TGW) of homozygous HJX74 (21.10 ± 0.07 g) was significantly different from that of 21 SSSLs (*p* < 0.05, *t*-test), and 13 QTLs for TGW were detected. Among these, four QTLs for TGW were distributed on chromosomes 1, 3, and 8, and on chromosomes 4, 6, 7, and 10, each contained three QTLs. S42 and S45 overlapped with S2, whilst S8 overlapped with S43 on chromosome 1. S40 overlapped with S46 and S48 on chromosome 3. S25 overlapped with S18 and P19 on chromosome 8. S27 overlapped with S35 and S47 on chromosome 7. Two overlaps were noted between S4, S23, and S24 on chromosome 10 (Figure 1).

Similarly, the GL of HJX74 (8.12 ± 0.35 mm) was significantly different from that of 14 SSSLs, and 11 QTLs for GL were detected. Moreover, the GW of HJX74 (2.52 ± 0.19 mm) was significantly different from that of 11 SSSLs, and nine QTLs for GW were detected. The LWR of HJX74 (3.24 ± 0.27) was significantly different from that of seven SSSLs, and six QTLs for LWR were detected. The GS of HJX74 (16.91 ± 1.52 mm^2^) was significantly different from that of 10 SSSLs, and seven QTLs for GS were detected. The GC of HJX74 (19.40 ± 0.77 mm) was significantly different from that of seven SSSLs, and six QTLs for GC were detected. The GR of HJX74 (0.33 ± 0.02) was significantly different from that of three SSSLs, and two QTLs for GR were detected. In total, 54 QTL loci were detected for seven grain shape traits, among which 20 exhibited positive effects. Specifically, 7, 6, 2, 2, 3, 2, and 1 QTLs exhibited positive effects for TGW, GL, GW, LWR, GS, GC, and GR, respectively (Table 1). Note that the overlapping QTL were counted as a single QTL.

### 2.4. Identification of Heterosis Loci

TGW of 15 SSSLs was significantly different from that of their respective F_1_ populations. Two of the three SSSLs overlapped on chromosome 1, and three of the four SSSLs overlapped on chromosome 10 (Figure 1). In total, 10 heterosis loci for TGW were detected in 15 SSSLs, among which six exhibited positive dominant effects. In addition, the GL of seven SSSLs was significantly different from that of their respective F_1_ populations. Three of the four SSSLs overlapped on chromosome 3. Six heterosis loci for GL were detected, among which five exhibited positive dominant effects. The GW of four SSSLs was significantly different from that of their respective F_1_ populations. The heterosis loci for GW were distributed on chromosomes 3, 7, 10, and 12, and two of them exhibited negative dominant effects. The LWR of seven SSSLs was significantly different from that of their respective F_1_ populations. Two of the three SSSLs overlapped on chromosome 3. Six heterosis loci for LWR were detected, among which five showed positive dominant effects (Table 2). In total, 26 heterosis loci for four grain shape traits, including 17 loci with the most stable positive genetic effects, were detected and are expected to improve rice yield and quality in rice breeding applications.

### 2.5. QTLs Linkage Analysis

QTL linkage analysis indicated that the interval PSM169–RM258 of S24 produced pleiotropic effects on six traits of grain shape, including GL, GW, LWR, GS, GC, and TGW (Table 3), contributing −18.49%, −42.62%, −9.18%, −38.32%, −23.44%, and −17.81% of the phenotypic variation in the early season of 2018 and −22.28%, −30.35%, −4.09%, −22.56%, −26.56%, and −26.78% of the phenotypic variation in the early season of 2019, respectively. Alleles from the receptor HJX74 increased the GL, GW, LWR, GS, GC, and GR, whereas parent S24 retained the QTL/HQTL for yield per plant (Table 4). In the present study, QTLs for grain shape were detected in the QTL-focus region of chromosome 3. The intervals RM633–RM16 of S40 and RM633–RM168 of S48 were detected at a similar locus as *GS3*.

Two QTLs for panicle length were identified in the intervals RM225–RM190 of S50 and RM505–RM234 of S17. Interestingly, these loci were also associated with grain shape (Table 1). The interval RM505–RM234 of S17 harboring grain shape QTLs and panicle shape QTLs was semblable to the semi-dominant gene *GW7* (*LOC_Os07g4120*), which was related to the production of more slender grains and the improvement of rice yield and grain quality [30]. Therefore, the grain shape genes had a pleiotropic effect, not only between grain shape traits but also with other yield-related traits.

### 2.6. Fine Mapping of QTLs for Grain Shape

SSSL-S24 was an exotic small-grain line compared with the receptor parent HJX74 (Table 1). The QTL identified in the interval RM271–PSM407 of S24 during the early and late seasons of 2016 might represent a new gene regulating grain shape. By mapping the population of sub-SSSLs and F_2_ populations, the interval range was narrowed from 2253.039 kb (PSM169–RM258) in 2018 to 75.124 kb (PSM169–RM25753) in 2019 (Figure 3), harbouring eight predicted genes (*LOC_Os10g37850.2*, *LOC_Os10g37860*, *LOC_Os10g37870*, *LOC_Os10g37880*, *LOC_Os10g37899*, *LOC_Os10g37920*, *LOC_Os10g37940*, and *LOC_Os10g37950*), among which novel QTL were contained for grain shape [40]. According to the RAP database (Rice Genome Annotation Project), the expression levels of the above four genes (*LOC_Os10g37850.2*, *LOC_Os10g37860*, *LOC_Os10g37870*, *LOC_Os10g37880*) were elevated in the seeds and panicles (Appendix A). Based on expression analysis, *LOC_Os10g37880* might be a candidate gene by expression primers (Figure 4) (Appendix A). Cloning this gene will offer deeper insights into the molecular breeding application of rice with lesser grain weight, higher grain yield, superior quality, and positive heterosis in the future.

### 2.7. QTLs Breeding Application for Grain Shape

In addition to the new QTLs for grain shape, S24 also carried QTLs for heading date and plant height. The yield of another substitution line, S29, was slightly higher than that of the control, but it did not carry QTLs for grain shape (Table 4). The use of this material was conducive to the study of the genetic breeding effect for the grain shape of S24. The yield of test crossing combination F_1_(HJX74/S24) was significantly higher than that of the control, while the GL, GW and TGW were significantly reduced. In addition, the heading date and plant height of F_1_(HJX74/S24) were the same as those of S24. The yield of the pyramiding line D7(S24/S29) was significantly higher than that of the control, and it also had the same QTLs for grain shape, heading date and plant height with S24. It should be noted that the difference in the number of panicles was not significant, but the TGW reduced significantly, indicating the yield was probably dominated by the significant increase in the number of grains per panicle (Figure 5).

## 3. Discussion

### 3.1. Breeding Applications of Grain Shape QTLs

Pursuing grain yield is one of the most critical objectives of rice breeding. The genetic bases of grain shape have received much attention because of their importance to rice yield and quality [2,28]. QTLs for grain shape have been widely studied in rice genetics. *GS3* harboured multiple alleles in the fifth exon, and each independent deletion variant caused a premature stop codon, conferring a short-seeded phenotype [31]. *GW7* was correlated with the production of more slender grain owing to increased cell division in the longitudinal direction and decreased cell division in the transverse direction in the gene-upregulated expression. Moreover, the *GW7^TFA^* allele in tropical *japonica* rice was associated with superior grain quality without a yield penalty, providing a new strategy by manipulating the *OsSPL16-GW7* module to simultaneously improve rice yield and grain quality [30]. Our study showed that S29 carried a yield QTL with a negative additive effect and a positive dominant effect on grain shape. The yield difference of S24 was not significant, but its dominance effect (F_1_) of yield QTL was positive and significant. The pyramiding line D7(S24/S29), which had the same QTL as S24 for small 1000-grain weight, high stalk, and late heading day, showed a significant increase in yield and the number of grains per panicle, while the number of panicles remained semblable and the 1000-grain weight decreased, indicating that the grain yield was dominated by the increase in the number of full grains per panicle. The loci RM505–RM234 and the adjacent RM633 identified in the present study were close to the previously reported QTL-focus regions harbouring genes *GS3* and *GW7* on chromosomes 3 and 7, respectively. Therefore, QTLs exist in hot mapping regions. In addition, pyramiding breeding of QTLs for grain shape played a pivotal role in obtaining rice with higher yield and superior quality [41]. Therefore, these QTLs for grain shape detected by the study would have ideal significance for the improvement of rice yield and quality in the future.

### 3.2. Pleiotropic Effects of QTLs for Grain Shape

Natural variations in the functions of genes encoding grain shape proteins result in pleiotropic effects of QTLs for grain shape and size in rice [26]. In rice, the appearance of grain shape is a result of the simultaneous regulation of multiple genes, which are either several tightly linked QTLs or pleiotropic QTLs with opposite effects on this trait. For the map-based cloning of such QTLs, interaction effects or non-genetic factors (i.e., epigenetic factors) must be removed [30,35]. According to the results, the grain shape genes also have a pleiotropic effect, not only between grain shape traits but also with panicle shape traits (Table 3).

In rice production, mono-effect QTLs can be easily used for rice improvement; however, the application of pleiotropic QTLs with opposite effects on different traits is complicated. In modern breeding, with the exception of yield, more attention should be paid to grain quality, which is mainly evaluated based on the appearance of the rice grain, as represented by the length-width-ratio (LWR) [29]; therefore, the manipulation of alleles for grain shape with pleiotropic effect is beneficial to rice yield and quality improvement. The results of this study also showed that higher plant height was not necessarily associated with longer grain shape, as in S24, which had higher plant height and produced shorter and smaller grains. Supposedly, small grains of rice may be an indicator of admirable rice quality. Nonetheless, whether QTLs for grain shape and panicle length produce pleiotropic effects warrants further research [42].

### 3.3. SSSL as an Excellent Material for QTL Mapping

Many QTLs are finally defined to a wide interval of approximately 10 cM in a primary mapping population because of the noise from their genetic backgrounds. Recent studies on QTL fine mapping have revealed that some tiny chromosome regions produce pleiotropic effects on TGW, SPP (spikelets per panicle) [43], and grain per the main panicle [29] and that their interactions are hardly removed in the general population. Owing to advanced mapping populations, such as CSSLs and SSSLs, the background noise can be greatly reduced, and fine mapping accuracy can be significantly improved. In these lines, regardless of the closely linked QTLs or pleiotropic QTLs in a small region, the target region harbouring genes may balance the allocation between grain weight and shape. Therefore, the same or similar intervals mapped in QTLs for several grain shape traits can be distinguished more easily in SSSLs [32].

## 4. Materials and Methods

### 4.1. Plant Materials

In 2016 and 2017, a population including 33 SSSLs derived from receptor HJX74(W0) and 8 donors (Appendix A) was selected to identify QTLs for grain shape in the Key Laboratory of Plant Molecular Breeding, Guangdong Province. All of the SSSLs were developed through three back-crossings followed by five to eight self-crossings and checked at each step by molecular marker-assisted selection. The distribution of each SSSL on the genetic linkage map is depicted in Figure 1.

### 4.2. Field Planting

Germinated seeds were sown on a seedling bed. Each line was transplanted with 30-day-old seedlings (late February) in the early season and 20-day-old seedlings (late July) in the late season. The seedlings were planted in two rows (with 10 plants per row) with 16.5 cm of distance between the plants and 30 cm of distance between the rows. Plantation was done following a randomised complete block design with two replicates on a homogenous field [35]. The plants were grown in paddy fields under natural long-day conditions (from March to July) during the early season and natural short-day conditions (from July to November) during the late season. The field was equipped with a bird net to prevent damage caused by birds.

### 4.3. Phenotypic Evaluation

Seeds collected (mid-July and late November) from primary panicles were dried for a week in a glasshouse. Fifty neat and evenly dried seeds from each line were evaluated per replicate. Six grain shape traits, including grain length (GL), grain width (GW), length-to-width ratio (LWR), grain size (GS), grain circumference (GC), and grain roundness (GR), were scanned and calculated using a Wan-Shen SC-A-type scanner (ScanMaker *i*800 plus); LWR was calculated as GL divided by GW, and the weight of 1000 grains was measured as 1000-grain weight (TGW). Ten plants per line from the middle of the two rows in each plot were harvested and measured, and the average of measurements was used as the phenotype of each line to determine the traits of grain shape per plant [14]. Panicle length was measured from the panicle neck section to the tip of the tallest panicle [32].

### 4.4. DNA Extraction and Genotyping

DNA was extracted using a micro-isolation method, as previously described [34]. Approximately 2 cm of young rice leaves at the four-leaf seedling stage were cut and ground in a GENO/TissueLyser-192 grinder for DNA extraction. Mini-scale DNA extraction was performed using the modified TE mixture. Genomic DNA used for analysing polymorphic markers was extracted using TPS solution (100 mM Tris–HCl (pH 8.0), 10 mM ethylenediaminetetraacetic acid (EDTA), and 1 M KCl).

The polymerase chain reaction (PCR) amplification conditions for all markers were the same as those previously described [36,37]. DNA fragments were amplified using PCR under the following thermal cycling conditions: denaturation for 5 min at 94 °C; followed by 38 cycles of denaturation for 30 s at 94 °C, annealing for 30 s at 55 °C, extension for 45 s at 72 °C; and a final extension for 5 min at 72 °C.

Each 15 μL reaction mixture contained 5.3 μL of master mix (Genstar 2× Taq PCR Star Mix with loading dye), 1.2 μL each of R/F primers (2–3 μM), and 2.0 μL of the extracted genomic DNA as the template, raised to 15 μL with water. The PCR products (approximately 5 μL) were separated by electrophoresis on 6% (*w*/*v*) non-denaturing polyacrylamide gel and subjected to silver staining as previously described [38].

All SSR markers were identified from the Gramene database (http://www.gramene.org/, accessed on 21 October 2013), and simple-sequence repeat (SSR) primers were designed according to the International Rice Genome Sequencing Project (http://rice.plantbiology.msu.edu/, accessed on 21 October 2013). A total of 560 SSR and InDel markers distributed throughout the genome were used to analyse the polymorphism and genetic background between the donor and receptor parents, among which 225 (40.18%) polymorphic markers were finally retained. These polymorphic markers were used to confirm the genotypes of the F_1_ and F_2_ lines, which were used as the source material to develop sub-SSSLs in this study.

### 4.5. Data Analysis

Phenotypic means were compared using ANOVA and Tukey’s multiple comparison tests with the Minitab software package (Release 13.1). The molecular linkage map was constructed using Mapmaker 3.0, setting the logarithm of odds (LOD) value to 3.0 [44]. The Kosambi function was used to calculate genetic distance. Composite interval mapping was performed for QTL analysis on the F_2_ populations using ICIM 4.1 (http://www.isbreeding.net/software/?type=detail&id=14, accessed on 21 October 2013). Marker intervals and the site at the peak value were determined, and candidate intervals were provided. The significance level to determine the candidate intervals and detect putative QTL and HQTL was set at a probability level of 0.01 and 0.05. A similar *p*-value was used to test the significance of the QTL effect. The length of the substitution segment was measured from sites a to d, and the estimated length of the substitution segment was measured from sites b to c in centimol units (Figure 6).
Add effect (A) = SSSL − W0
Add effect variation explained (APVE, %) = (SSSL − W0)/W0 × 100
D effect (D) = F1 − (SSSL + W0)/2
D effect variation explained (DPVE, %) = [D/(SSSL + W0)/2] × 100.

### 4.6. Fine Mapping and Expression Analysis

During the early and late seasons of 2018 and the early season of 2019, S17, S24, S40, S48, and S50 were used for further QTL linkage analysis on a continuous F_2_ population for finer mapping with 280 sequentially developed plants. Of the 280 plants, 224 were subjected to linkage analysis using novel polymorphic molecular markers. The expression of candidate genes was performed as previously described [2].

## 5. Conclusions

A SSSL with a single donor fragment and chromosomal region harbouring the target trait may be considered a QTL. In this study, 54 QTLs for grain shape were detected, including some QTLs reported previously, such as *GS3*, *GW8*, and *GW7*, indicating the feasibility of identifying QTLs using SSSLs. In addition, a novel QTL *qTGW10-3* for grain shape was identified in the interval RM271–PSM407 of S24. Linkage mapping revealed that the interval range of the target trait could be narrowed down from 2253.039 kb (PSM169–RM258) to 75.124 kb (PSM169–RM25753), and the candidate gene was identified as *LOC_Os10g37880* which could be applied to breeding practice by molecular marker-assisted selection. Therefore, QTL identification using SSSL is feasible, convenient, and rapid. Amongst the detected QTLs, *qTGW4-1*, *qTGW4-2* (PSM106–PSM382; RM348–RM127), *qTGW5-1* (PSM202–RM437), *qTGW7-1* (PSM142–RM70), and *qTGW10-2* (END–RM596) showed a high positive-additive genetic effect on grain shape. *qTGW10-3* (RM271-PSM407) showed a high dominant genetic effect on grain yield. In the future, these novel genes might be cloned and used in breeding to improve grain yield and the quality of rice.

## Figures and Tables

**Figure 1 plants-12-00892-f001:**
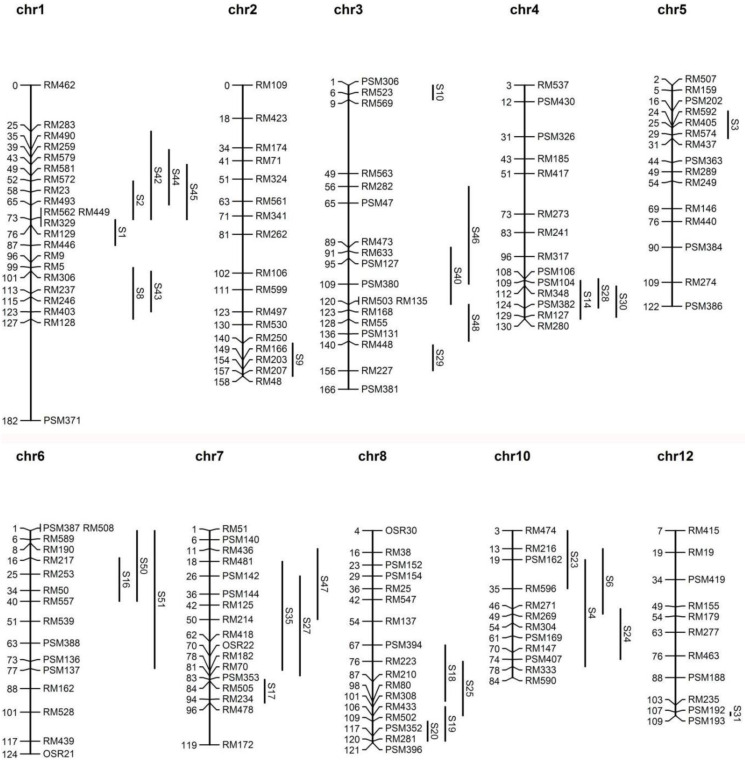
Distribution of the 33 single-segment substitution lines in chromosomal marker intervals. Substituted segments are represented by black lines. Symbols on the right of each segment indicate the abbreviations of the experimental materials. Codes on the left and right sides of each chromosome indicate the designated centimorgan (cM) and molecular markers, respectively.

**Figure 2 plants-12-00892-f002:**
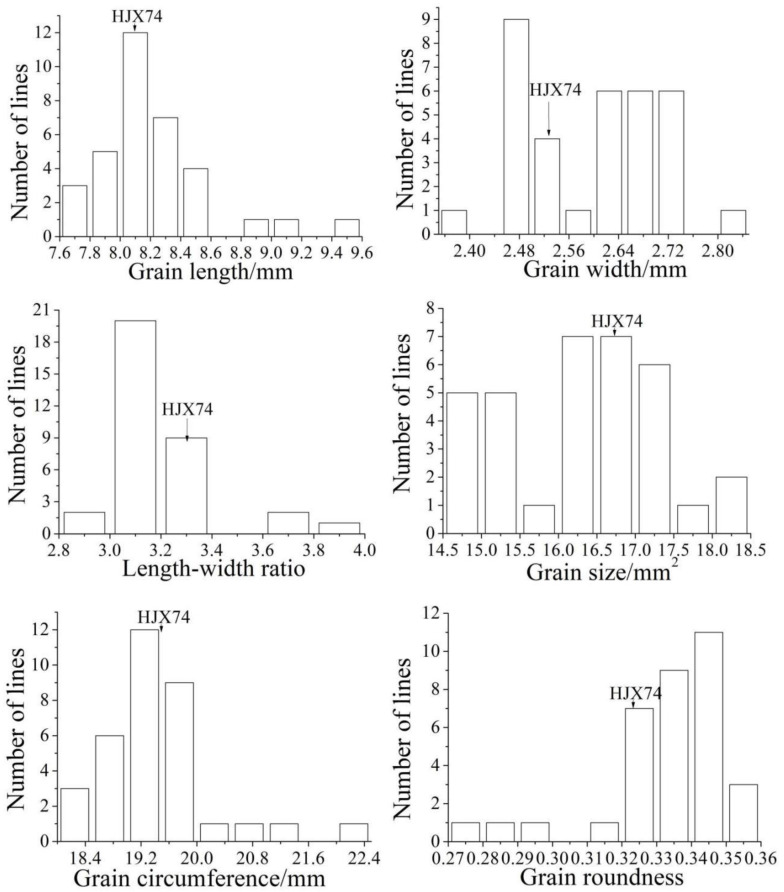
Frequency distribution of grain shape traits in SSSLs. Phenotypes of the receptor parent (HJX74) are indicated by arrows. Data are presented as the mean of 10 plants of each line in independent seasons.

**Figure 3 plants-12-00892-f003:**
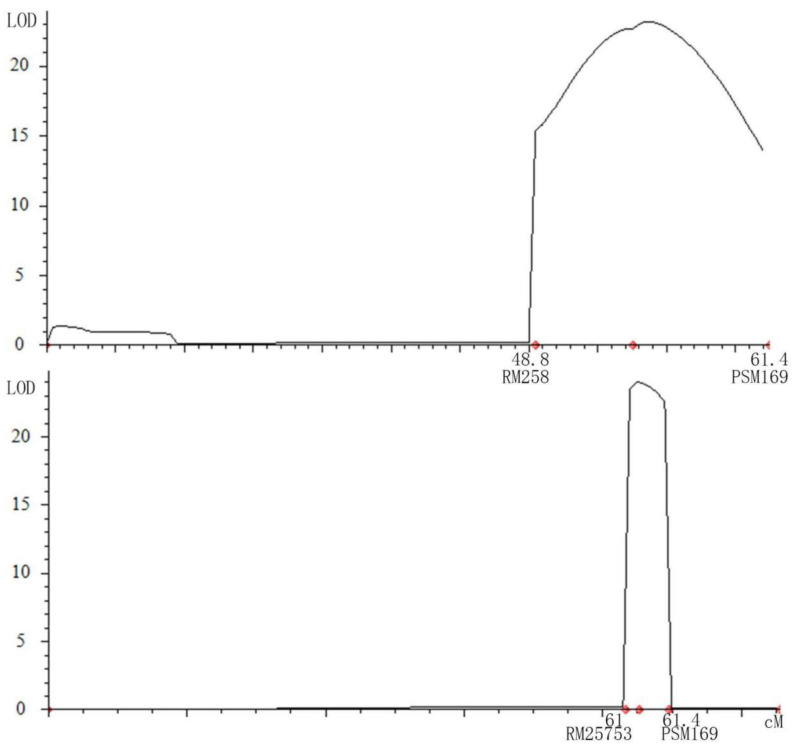
Putative location of the QTL for grain shape in S24. (**Up**) Mapped in 2018; (**Down**) mapped in 2019. Genetic distance (in cM, centimorgans) between two molecular markers is indicated on the *X*-axis. LOD is abbreviation of the logarithm/likelihood of odds on the *Y*-axis. The red symbols indicate marker sites or peak locations.

**Figure 4 plants-12-00892-f004:**
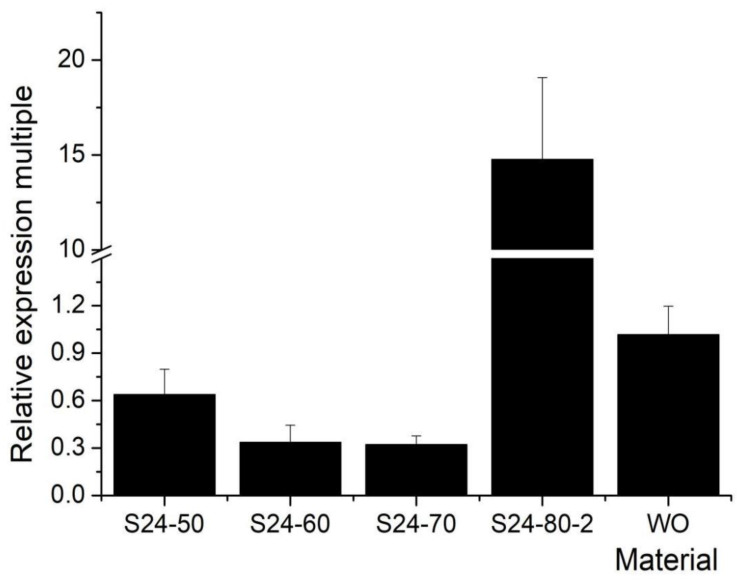
Expression pattern of *LOC_0s10g37880*. The *Y*-axis indicates the relative expression multiple, and the *X*-axis indicates the materials. W0 is short for HJX74 as the corresponding control. S24-50, S24-60, S24-70 and S24-80-2 represent the expression of genes *LOC_Os10g37850.2*, *LOC_Os10g37860*, *LOC_Os10g37870*, and *LOC_Os10g37880* in S24, respectively.

**Figure 5 plants-12-00892-f005:**
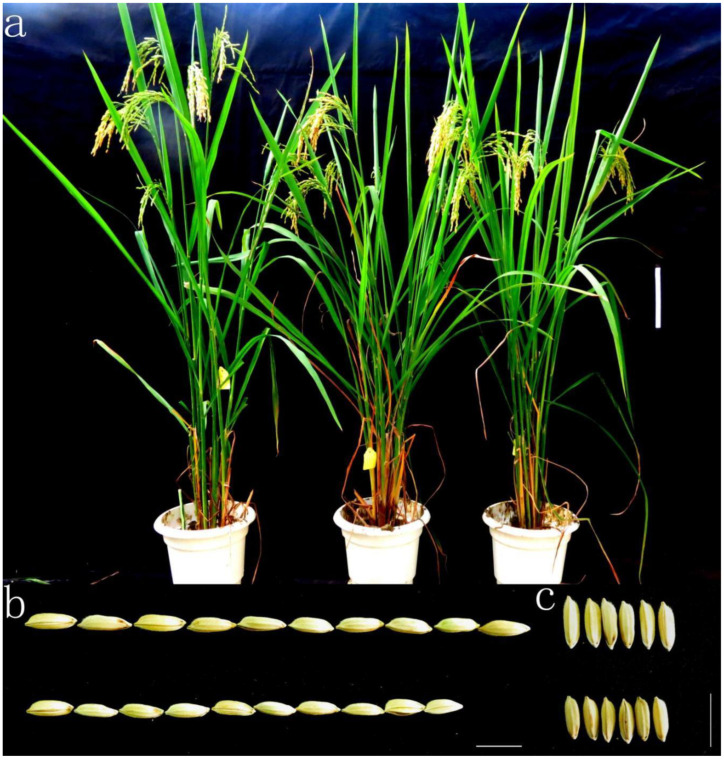
Plant appearance and grain shape of the pyramid line. (**a**) Plant appearance, from left to right: S24, D7 (S24/S29), HJX74. Bar = 15 cm. (**b**) Grain length for HJX74 (**up**) and S24 (**down**). Bar = 1 cm. (**c**) Grain width for HJX74 (**up**) and S24 (**down**). Bar = 1 cm.

**Figure 6 plants-12-00892-f006:**
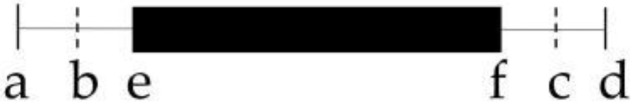
The measuring illustration of the length of the substitution fragment. (b and c were the median values of a and e, f and d, respectively).

**Table 1 plants-12-00892-t001:** QTLs Identified for Grain Shape Using SSSLs (n = 10).

Trait	Material	QTL	Marker Interval	Mean ± SE	Add Effect	APVE%	*p*-Value
TGW/g	S1	*qTGW1−1*	RM329–RM9	20.80 ± 0.01	−0.0900	−2.12	0.004
	S8	*qTGW1−2*	RM5–RM128	20.50 ± 0.02	−0.1500	−3.53	<0.001
	S42	*qTGW1−3*	RM283–RM562	22.50 ± 0.03	0.1600	3.76	<0.001
	S43	*qTGW1−4*	RM306–RM403	22.25 ± 0.02	0.2000	4.71	<0.001
	S10	*qTGW3−1*	END–RM569	19.35 ± 0.03	−0.3800	−8.94	<0.001
	S40	*qTGW3−2*	PSM126–RM503	23.55 ± 0.01	0.4884	11.58	<0.001
	S46	*qTGW3−3*	PSM16–PSM128	21.75 ± 0.04	0.1000	2.35	<0.001
	S48	*qTGW3−4*	RM135–RM448	24.40 ± 0.02	0.6300	14.82	<0.001
	S28	*qTGW4−1*	PSM106–PSM382	22.95 ± 0.01	0.3691	8.75	<0.001
	S30	*qTGW4−2*	RM348–RM127	23.60 ± 0.02	0.4700	11.06	<0.001
	S3	*qTGW5−1*	PSM202–RM437	22.40 ± 0.03	0.2651	6.29	<0.001
	S50	*qTGW6−1*	RM135–RM448	19.24 ± 0.16	−0.5700	−2.88	0.0450
	S27	*qTGW7−1*	PSM142–RM70	22.20 ± 0.02	0.2188	5.19	<0.001
	S35	*qTGW7−2*	RM481–RM182	20.15 ± 0.02	−0.1863	−4.42	<0.001
	S47	*qTGW7−3*	PSM140–RM125	18.65 ± 0.02	−0.5200	−12.24	<0.001
	S18	*qTGW8−1*	PSM394–RM80	22.10 ± 0.02	0.1700	4.00	<0.001
	S19	*qTGW8−2*	RM308–RM281	20.45 ± 0.01	−0.1289	−3.06	<0.001
	S25	*qTGW8−3*	RM223–RM433	19.60 ± 0.03	−0.3300	−7.76	<0.001
	S4	*qTGW10−1*	PSM162–PSM163	20.25 ± 0.01	−0.2000	−4.71	<0.001
	S23	*qTGW10−2*	END–RM596	22.75 ± 0.02	0.2200	5.18	<0.001
	S24	*qTGW10−3*	RM271–PSM407	19.20 ± 0.01	−0.4100	−9.65	<0.001
GL/mm	S44	*qGL1−1*	RM490–RM562	8.00 ± 0.11	−0.1235	−1.52	0.0023
	S9	*qGL2−1*	RM425–END	8.49 ± 0.11	0.3715	4.57	<0.001
	S10	*qGL3−1*	END–RM569	7.55 ± 0.10	−0.5110	−6.34	<0.001
	S40	*qGL3−2*	PSM126–RM503	8.98 ± 0.08	1.0200	12.81	<0.001
	S48	*qGL3−3*	RM135–RM448	9.48 ± 0.15	1.3565	16.70	<0.001
	S28	*qGL4−1*	PSM106–PSM382	8.36 ± 0.13	0.2405	2.96	<0.001
	S14	*qGL4−2*	PSM104–END	8.52 ± 0.02	0.0600	0.71	<0.001
	S17	*qGL7−1*	PSM353–RM478	8.78 ± 0.11	0.7189	8.92	<0.001
	S35	*qGL7−2*	RM481–RM182	7.54 ± 0.10	−0.5240	−6.50	<0.001
	S47	*qGL7−3*	PSM140–RM125	7.35 ± 0.15	−0.7160	−8.88	0.0029
	S25	*qGL8−1*	RM223–RM433	7.79 ± 0.11	−0.2780	−3.45	0.0421
	S23	*qGL10−1*	END–RM596	8.27 ± 0.10	0.2088	2.59	<0.001
	S24	*qGL10−2*	RM271–PSM407	7.45 ± 0.11	−0.6150	−7.63	<0.001
	S31	*qGL12−1*	PSM192–PSM193	7.96 ± 0.03	−0.1100	−1.36	<0.001
GW/mm	S42	*qGW1−1*	RM283–RM562	2.65 ± 0.01	0.0300	1.15	0.0370
	S45	*qGW1−2*	RM579–RM594	2.50 ± 0.03	−0.0390	−1.54	0.0199
	S10	*qGW3−1*	END–RM569	2.41 ± 0.05	−0.0700	−3.00	0.0257
	S40	*qGW3−2*	PSM126–RM503	2.34 ± 0.06	−0.1980	−7.80	0.0047
	S48	*qGW3−3*	RM135–RM448	2.46 ± 0.05	−0.0823	−3.24	<0.001
	S50	*qGW6−1*	PSM387–RM557	2.43 ± 0.05	−0.1060	−4.17	<0.001
	S17	*qGW7−1*	PSM353–RM478	2.34 ± 0.06	−0.2002	−7.88	<0.001
	S47	*qGW7−2*	PSM140–RM125	2.48 ± 0.05	−0.0623	−2.47	0.0044
	S25	*qGW8−1*	RM223–RM433	2.39 ± 0.01	−0.0400	−1.65	0.0170
	S24	*qGW10−1*	RM271–PSM407	2.45 ± 0.02	−0.0800	−3.16	<0.001
	S31	*qGW12−1*	PSM192–PSM193	2.72 ± 0.01	0.0400	1.49	<0.001
LWR	S8	*qLWR1−1*	RM5–RM128	3.02 ± 0.08	−0.1533	−4.83	0.0077
	S44	*qLWR1−2*	RM490–RM562	2.96 ± 0.06	−0.2150	−6.77	0.0153
	S40	*qLWR3−1*	PSM126–RM503	3.64 ± 0.11	0.4650	14.64	<0.001
	S48	*qLWR3−2*	RM135–RM448	3.88 ± 0.10	0.6330	19.51	<0.001
	S17	*qLWR7−1*	PSM353–RM478	3.78 ± 0.10	0.6009	18.71	<0.001
	S47	*qLWR7−2*	PSM140–RM125	2.96 ± 0.08	−0.2200	−6.92	0.0237
	S31	*qLWR12−1*	PSM192–PSM193	2.93 ± 0.02	−0.0900	−2.98	<0.001
GS/m2	S42	*qS1−1*	RM283–RM562	16.75 ± 0.12	0.3000	1.82	0.0050
	S45	*qS1−2*	RM579–RM594	14.27 ± 0.30	−1.8300	−11.37	0.0012
	S40	*qS3−1*	RM223–RM433	17.52 ± 0.18	1.6100	10.12	<0.001
	S48	*qS3−2*	RM135–RM448	17.91 ± 0.16	1.9100	11.94	<0.001
	S50	*qS6−1*	PSM387–RM557	16.14 ± 0.50	−0.7710	−4.56	0.0137
	S51	*qS6−2*	END–PSM137	15.15 ± 0.42	−1.7610	−10.42	0.0332
	S47	*qS7−1*	PSM140–RM125	14.63 ± 0.43	−2.2750	−13.46	<0.001
	S25	*qS8−1*	RM223–RM433	16.19 ± 0.10	−0.3000	−1.82	<0.001
	S24	*qS10−1*	RM271–PSM407	15.46 ± 0.42	−1.4420	−8.53	0.0194
	S23	*qS10−2*	END–RM596	17.19 ± 0.21	0.4500	2.69	0.0040
GC/mm	S45	qC1−1	RM579–RM594	18.60 ± 0.23	−0.8060	−4.15	0.0132
	S40	*qC3−1*	PSM126–RM503	20.70 ± 0.34	1.2930	6.66	0.0174
	S48	*qC3−2*	RM135–RM448	21.46 ± 0.15	2.3900	12.53	<0.001
	S17	*qC7−1*	PSM353–RM478	21.27 ± 0.38	1.8710	9.64	<0.001
	S47	*qC7−2*	PSM140–RM125	17.29 ± 0.36	−1.3830	−7.41	0.0042
	S24	*qC10−1*	RM271–PSM407	17.21 ± 0.25	−1.4630	−7.84	<0.001
	S31	*qC12−1*	RM271–PSM407	19.06 ± 0.06	−0.2700	−1.40	<0.001
GR	S40	*qR3−1*	PSM126–RM503	0.28 ± 0.01	−0.0480	−14.50	<0.001
	S48	*qR3−2*	RM135–RM448	0.27 ± 0.01	−0.0550	−16.62	<0.001
	S31	*qR12−1*	PSM192–PSM193	0.35 ± 0.00	0.0100	2.94	<0.001

Notes: Add effect, additive effect; positive additive indicates SSSL alleles increasing the phenotypic value of grain shape. APVE%, add effect variation explained by the QTLs (i.e., the QTL was repeatedly detected in SSSL populations). The *p*-value indicates the significance level. All data are presented as mean ± SE. The *p*-values for each trait between HJX74 and SSSLs are obtained using a *t*-test. n represents the number of samples per SSSL material.

**Table 2 plants-12-00892-t002:** Effects of heterosis loci on MPH/dominance for grain shape (n = 10).

Trait	Material	QTL	Marker Interval	P	F1	A	APVE%	D	DPVE%	*p*-Value
TGW/g	S1	*qHTGW1−1*	RM329–RM9	20.60	4.24	−0.1300	−3.60	0.0550	1.31	<0.001
	S2	*qHTGW1−2*	RM572–RM449	21.35	4.42	0.0200	0.47	0.1600	3.76	<0.001
	S8	*qHTGW1−3*	RM5–RM128	20.50	4.29	−0.1500	−3.53	0.1150	2.75	<0.001
	S40	*qHTGW3−1*	PSM126–RM503	24.00	4.46	0.5580	13.15	−0.0610	−1.35	<0.001
	S48	*qHTGW3−2*	RM135–RM448	25.30	4.63	0.8180	19.28	−0.0210	−0.45	<0.001
	S28	*qHTGW4−1*	PSM106–PSM382	22.85	4.43	0.3524	8.36	0.0362	0.82	<0.001
	S30	*qHTGW4−2*	RM348–RM127	23.60	4.41	0.4700	11.06	0.0750	1.67	<0.001
	S17	*qHTGW7−1*	PSM353–RM478	21.10	4.31	0.0024	0.06	0.0912	2.16	0.0034
	S20	*qHTGW8−1*	RM502–RM281	21.15	4.28	−0.0200	−0.47	0.0400	0.94	<0.001
	S25	*qHTGW8−2*	RM223–RM433	19.85	4.08	−0.3800	−8.94	−0.0200	0.49	<0.001
	S4	*qHTGW10−1*	PSM162–PSM163	20.10	4.24	−0.2000	−4.71	0.0900	2.17	<0.001
	S6	*qHTGW10−2*	RM216–RM269	20.75	4.27	−0.1000	−2.35	0.0700	1.67	<0.001
	S23	*qHTGW10−3*	END–RM596	22.25	4.35	0.2000	4.71	0.0000	0.00	<0.001
	S24	*qHTGW10−4*	RM271–PSM407	19.40	3.98	−0.3700	−8.71	−0.0850	−2.09	<0.001
	S31	*qHTGW12−1*	PSM192–PSM193	21.45	4.24	0.0724	1.72	−0.0138	−0.32	0.0201
GL/mm	S29	*qHGL3−1*	RM293–RM227	7.81	8.02	−0.3105	−3.82	0.0548	0.69	0.0083
	S40	*qHPL3−2*	PSM126–RM503	8.98	8.17	1.0200	12.81	−0.3000	−3.54	<0.001
	S48	*qHPL3−3*	RM135–RM448	9.13	8.18	1.1100	13.84	−0.3950	−4.61	<0.001
	S17	*qHGL7−1*	PSM353–RM478	8.78	8.45	0.7170	8.89	0.0285	0.34	<0.001
	S35	*qHGL7−2*	RM481–RM182	7.54	7.83	−0.5230	−6.49	0.0285	0.37	<0.001
	S24	*qHGL10−1*	RM271–PSM407	7.55	7.83	−0.5130	−6.36	0.0235	0.30	<0.001
	S31	*qHGL12−1*	PSM192–PSM193	7.74	7.91	−0.3230	−4.01	0.0085	0.11	0.0090
GW/mm	S48	*qGW3−1*	RM135–RM448	2.33	2.50	−0.1680	−6.72	0.0830	3.44	0.0036
	S17	*qHGW7−1*	PSM353–RM478	2.22	2.25	−0.0700	−3.06	−0.0050	−0.22	<0.001
	S24	*qHGW10−1*	RM271–PSM407	2.45	2.49	−0.0800	−3.16	0.0000	0.00	<0.001
	S31	*qHGW12−1*	PSM192–PSM193	2.61	2.58	0.0500	1.95	−0.0050	−0.19	0.0370
LWR	S29	*qHLWR3−1*	RM293–RM227	2.98	3.09	−0.1970	−6.20	0.0115	0.37	0.0265
	S40	*qLWR3−2*	PSM126–RM503	3.88	3.31	0.6990	21.97	−0.2245	−6.36	0.0021
	S48	*qLWR3−3*	RM135–RM448	3.97	3.38	0.7870	24.74	−0.1975	−5.53	<0.001
	S16	*qHLWR6−1*	RM217–RM557	2.97	3.13	−0.2070	−6.52	0.0565	1.84	0.0037
	S17	*qHLWR7−1*	PSM353–RM478	3.78	3.50	0.6030	18.98	0.0215	0.62	<0.001
	S24	*qHLWR10−1*	RM271–PSM407	3.13	3.26	−0.0470	−1.48	0.1065	3.38	0.0190
	S31	*qHLWR12−1*	PSM192–PSM193	2.92	3.08	−0.2570	−8.09	0.0315	1.03	<0.001

Notes: P, parent; D, dominance effect; positive dominance indicated that the hybrid values are higher than the mid-parent values. F_1_: The hybrids derived from the SSSLs and the receptor parent HJX74. MPH: mid-parent heterosis. n represents the number of samples per SSSL material.

**Table 3 plants-12-00892-t003:** QTL linkage analysis of important grain shape traits (n = 224).

Material	Trait	Marker Interval	2018E		2018L		Marker	2019E	
			LOD	PVE (%)	LOD	PVE (%)		LOD	PVE (%)
S24	GL	PSM169–RM258	8.71	−18.49	17.36	−38.00	PSM169–RM25753	16.86	−22.28
	GW	PSM169–RM258	23.15	−42.62	8.22	−21.89	PSM169–RM25753	24.00	−30.35
	LWR	PSM169–RM258	3.95	−9.18			PSM169–RM258	2.77	−4.09
	GS	PSM169–RM258	20.38	−38.32	19.71	−42.31	PSM169–RM25753	16.92	−22.56
	GC	PSM169–RM258	11.46	−23.44	18.79	−40.33	PSM169–RM25753	20.61	−26.56
	TGW	PSM169–RM258	9.80	−17.81	7.67	−18.93	RM25753–RM147	20.24	−26.78
S40	GL	RM633–RM16	77.51	73.33	81.82	84.11			
	GW	RM633–RM16	18.51	34.64	13.64	30.50			
	LWR	RM633–RM16	70.68	72.21	85.70	86.54			
	GS	RM633–RM16	43.52	61.44	37.97	61.76			
	GC	RM633–RM16	79.07	73.65	76.19	83.10			
	GR	RM633–RM16	66.43	71.60	74.37	83.43			
	TGW	RM633–RM16	25.10	44.16	9.35	22.67			
S48	GL	RM633–RM168	99.66	46.78	80.12	75.55			
	GW	RM633–RM168	11.16	22.36	9.88	23.51			
	LWR	RM633–RM168	74.75	40.38	74.05	71.12			
	GS	RM633–RM168	47.23	68.16	34.37	57.96			
	GC	RM633–RM168	95.79	46.55	77.34	75.73			
	GR	RM633–RM168	64.210	37.84	68.53	70.07			
	TGW	RM633–RM168	34.65	30.40	14.90	28.73			
S50	MPL	RM225–RM190	2.72	6.31			RM190–RM557	7.59	5.67
	VPL	RM225–RM190	6.11	11.54			RM225–RM190	8.58	8.00
	TGW	RM50–RM557	3.76	3.08			RM225–RM190	6.42	6.52
	GW						RM190–RM557	3.35	3.16
	LWR						RM225–RM190	2.93	3.06
	GC						RM190–RM557	2.99	0.00
S17	GL						RM505–RM234	49.32	52.28
	GW						RM478–RM505	8.38	11.61
	LWR						RM478–RM505	99.83	37.58
	GC						RM505–RM234	27.27	33.92
	TGW						RM478–RM505	4.95	1.05
	VPL						RM505–RM234	10.55	14.39
	MPL						RM505–RM234	6.94	9.79

Notes: GL, grain length; GW, grain width; LWR, length-to-width ratio; GS, grain size; GC, grain circumference; GR, grain roundness; TGW, 1000-grain weight; VPL, average panicle length; MPL, maximum panicle length; E, early season; L, late season. LOD values indicate significance levels. n represents the number of plants in F_2_. Population.

**Table 4 plants-12-00892-t004:** Phenotypes of QTLs for grain shape and yield.

Materials	Yield per Plant	Panicle Number	The Number of Full Grains	Thousand-Grain Weight	Panicle Length	Grain Length	Grain Width	Heading Day	Plant Height
HJX74	22.4	6.8	1087.2	21.10	22.4	8.130	2.530	97.0	105.2
S24	21.5	6.8	1166.8	18.50 **	21.9	7.584 *	2.402 *	103.0 *	119.2 *
S29	23.0 *	6.4	1078.6	21.55	21.1	7.813	2.559	98.0	106.4
F_1_ (HJX74/S24)	26.1 **	7.7	1312.2 **	20.02 *	22.2	7.775 *	2.407 *	100.0 *	116.1 *
D7 (S24/S29)	23.8 *	6.7	1254.1 *	18.76 *	21.8	7.673 *	2.408 *	103.0 *	110.1 *

* and ** denote the significance levels at *p* < 0.05 and *p* < 0.01, respectively.

## Data Availability

All the data are available from the corresponding author upon reasonable request.

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
