# Peer review of "Identification, Fine Mapping and Application of Quantitative Trait Loci for Grain Shape Using Single-Segment Substitution Lines in Rice (*Oryza sativa* L.)"

_plants, 2023, doi:10.3390/plants12040892_

Round 1

Reviewer 1 Report

Dear Editor,

In the paper entitled, “Identification and application of quantitative trait loci for grain shape using single segment substitution lines in rice (Oryza sativa L.)”, the authors have identified many QTLs related to grain shape using SSSL’s of rice. In addition, the authors have identified the candidate genes related to grain shape by fine mapping the predicted QTL region. The authors need to do major changes to the text in Introduction, materials and methods, Results and Discussion parts. In addition, all the figures and tables description need to be specified in detail to make the readers understand the data. Furthermore, the authors need to carry the following corrections to the manuscript.

1.      In this study the authors have used rice SSSL lines for the identification of QTLs, so it is suggested to include the importance of Single Segment Substitution Lines in Introduction.

2.      In results section 2.1, the authors specified they have used 225 SSRs for genotyping but, they have not included the details. The authors need to include the details of SSRs as supplementary table to the manuscript.

3.      The abbreviations in the manuscript can be fully explained at first time use in the manuscript.

4.      As described in the above text the descriptions of all tables and figures should be explained in detail.

5.      In materials and methods section, the details of plant materials are less informative and need to be revised.

6.       Also, in materials and methods, section 4.4 title can be changed to “DNA extraction and genotyping” and title 4.5 as “Data analysis”.

Author Response

Dear reviewer:

The major changes to the text in Introduction, Materials and Methods, Results and Discussion parts have been made suitably following your good suggestions, and all the figures and tables description have been specified in more detail.

  1. The importance of Single Segment Substitution Lines has been includedin the  Introduction.
  2. The details of SSR has been providedas Supplementary table.1 .
  3. The abbreviations in the manuscript has beenfully explained at first time use in the manuscript.
  4. All tables and figures have been explained in more
  5. The detailed informationof plant materials has been revised in materials and methods section.
  6. Section 4.4 title has beenchanged to “DNA extraction and genotyping” and section title 4.5 to “Data analysis” in materials and methods.

yours sincerely

Xiaoling Wang

February 8, 2023

Reviewer 2 Report

The manuscript presented deals with the identification and application of quantitative trait loci for grain shape in rice. As this is an important trait related to consumer preferences and the cooking quality of the rice grain the authors are tackling an interesting problem in this crop's production.

I have the following comments and questions for the authors:

While the abbreviations used for denoting various rice grain properties (i.e. GL, GW, LWR, GS, GC, and GR) are very much familiar to the cereals breeders, this is not the case for the broader readership. Therefore, the normal practice of writing the terms used in full at the first appearance in the text and using that to introduce the respective abbreviations has to be applied to this manuscript as well.

In Lines 66-67 the authors state that "the length of the substituted segments varied from 1.6–87.5 cM". This is immediately followed by the statement that "the estimated length was between 0.8–77.3 cM". This makes the two statements contradictory, which needs to be clarified/corrected.

In Lines 91-92 authors state that "three QTLs were distributed on chromosomes 4, 6, 7, and 10". This can not be correct for obvious reasons and needs to be clarified.

Figure 4 contains material denoted as "WO" which is not clarified in the caption or the nearby text. This needs to be corrected.

The English language needs checking for spelling errors, correcting sequence of tenses, the usage of single/plural forms of words and terms (i.e. QTLs should be used on multiple occasions instead of QTL), and even the capitalization of the first letters of the sentences. In some cases, incorrect words appear to have replaced established terms (i.e. mark instead of markers in Line 65, which seems to be a technical error in this particular case, but such ones should be cleaned throughout the text).

Author Response

Dear reviewer:

The major changes to the text in Introduction, Materials and Methods, Results and Discussion parts have been made suitably following your good suggestions, and all the figures and tables description have been specified in more detail.

  1. The importance of Single Segment Substitution Lines has been includedin the  Introduction.
  2. The details of SSR has been providedas Supplementary table.1 .
  3. The abbreviations in the manuscript has beenfully explained at first time use in the manuscript.
  4. All tables and figures have been explained in more
  5. The detailed informationof plant materials has been revised in materials and methods section.
  6. Section 4.4 title has beenchanged to “DNA extraction and genotyping” and section title 4.5 to “Data analysis” in materials and methods.

yours sincerely

Xiaoling Wang

February 8, 2023

Dear reviewer:

Thank you for your review. Modifications have been done following your good suggestions:

  1. The abbreviations of the terms used for denoting various rice grain properties (i.e. GL, GW, LWR, GS, GC, GR andTGW) have been written in full at the first appearance in the text.
  2. In added 4.5 Data analysis,the length of the substitution segment was measured from sites a to d, and the estimated length of the substitution segment was measured from sites b to c in cM units (b and c were the median values of a and e, f and d, respectively).
  3. S27 overlapped with S35 and S47 on chromosome 7. Two overlaps were noted between S4, S23, and S24 on chromosome 10 (Fig. 1). "Three QTLswere distributed on chromosomes 4, 6, 7, and 10 " has been revised as "on chromosomes 4, 6, 7, and 10, each contained three QTLs".
  4. "W0" has beenclarified as HJX74 in the annotation of Figure 4.
  5. The English language has been checkedfor spelling errors, sequence of tenses, the usage of single/plural forms of words and terms, and the capitalization of the first letters of the sentences.

yours sincerely

Xiaoling Wang

February 8, 2023

Round 2

Reviewer 1 Report

Dear Dr. Wang,

I would like to strongly suggest to change the title of manuscript and in addition, you need to go through the manuscript thoroughly for grammatical errors.

 Suggestion for the title:

1.      Identification and Fine mapping of quantitative trait loci for grain shape using single -segment substitution lines in rice (Oryza sativa L).

)

Author Response

Dear editors and referees,

The modification to the text in Abstract, Introduction, Materials and Methods, Results and Discussion parts have been made suitably and all the figures and tables description have been specified in more detail following your good suggestions.

1.The title of manuscript was changed to “Identification, fine mapping and application of quantitative trait loci for grain shape using single-segment substitution lines in rice (Oryza sativa L).

2.The number of SSR and IndDel primers was modified from 255 to 225 pairs.

3.The estimated length was removed because of not easy to describe.

4.The abbreviations (i.e. GL, GW, LWR, GS, GC, GR and TGW) in the manuscript have been fully explained at first time used in the manuscript.

5."Three QTLs were distributed on chromosomes 4, 6, 7, and 10 " has been revised as "on chromosomes 4, 6, 7, and 10, each contained three QTLs".

6.The format of the reference (Wang et al. 2015) was changed to [30].

7.Four candidate genes were specified explicitly as LOC_Os10g37850.2, LOC_Os10g37860, LOC_Os10g37870, LOC_Os10g37880.

8.Section 3.1 ”showed that the yield increased significantly, the number of panicle remained semblable, the 1000-grain weight decreased, and the number of grain per panicle increased significantly, with the same QTL of S24 for small 1000-grain weight, high stalk and late heading day, showing that the grain yield was dominated by the increase of the number of full gain per panicle.” were replaced with “which had the same QTL with S24 for small 1000-grain weight, high stalk and late heading day, showed significant increase in yield and the number of grain per panicle while the number of panicle remained semblable and the 1000-grain weight decreased, indicating that the grain yield was dominated by the increase of the number of full grain per panicle.”

9.The detailed information of plant materials has been revised in materials and methods section.

10.Section 4.4 title has been changed to “DNA extraction and genotyping” and section title 4.5 to “Data analysis” in materials and methods.

11.”The length of the substitution segment was measured from sites a to d, and the estimated length of the substitution segment was measured from sites b to c in centimol units (b and c were the median values of a and e, f and d, respectively).” was added to the Section 4.5.

12.Fund ” National Natural Science Foundation of China (no. 32060475)” was adjusted to first, and “Jiangxi seed industry joint research project-Excellent germplasm creation of japonica and cultivation of new varieties of indica-japonica hybrid rice” was added to the end.

13.A new reference [39] was inserted and all references were relevant to the contents of the manuscript.

14.The English language has been checked for spelling errors, sequence of tenses, the usage of single/plural forms of words and terms, and the capitalization of the first letters of the sentences by a native English-speaking colleague.

15.The details of SSR and inDel markers have been provided as Supplementary table.1 .

yours sincerely

Yulong Xiao

Xiaoling Wang

February 12, 2023
